# Prevalence and Characteristics of Australians Complementary Medicine Product Use, and Concurrent Use with Prescription and Over-the-Counter Medications—A Cross Sectional Study

**DOI:** 10.3390/nu15020327

**Published:** 2023-01-09

**Authors:** Joanna Harnett, Erica McIntyre, Jon Adams, Tamia Addison, Holly Bannerman, Lucy Egelton, Jessica Ma, Leon Zabakly, Amie Steel

**Affiliations:** 1Faculty of Medicine, Health School of Pharmacy, The University of Sydney, Camperdown, NSW 2006, Australia; 2Institute for Sustainable Futures, University of Technology Sydney, Ultimo, NSW 2007, Australia; 3Research Institute for Innovative Solutions for Wellbeing and Health, Faculty of Health, University of Technology Sydney, Ultimo, NSW 2007, Australia; 4ARCCIM, School of Public Health, University of Technology Sydney, Ultimo, NSW 2007, Australia

**Keywords:** complementary medicines, drug interactions, public health

## Abstract

Data about the characteristics and prevalence of complementary medicine (CM) product use by Australians, including concurrent use with prescription and over-the-counter medications, have not been collected in the last five years. A cross-sectional online survey involving a representative sample of the Australian population was administered in 2021–2022. Of the 2351 survey responses included in this study, 49.4% reported use of a CM product over the previous 12-month period. Of these, 50% reported they always or often used CM products on the same day as a prescription medicine. Participants aged 65 and over were five times more likely to use CMs and other medications on the same day compared to 18–24-year-olds. Lower levels of education and having a chronic illness were also predictors of same-day use. The prevalence and characteristics of CM use by participants was similar to data collected five years ago. The study shows that concurrent use of CM products with prescription medications among older and more vulnerable populations is prevalent and this area requires further research to help ensure appropriate and safe use of CM products.

## 1. Introduction

Complementary medicines (CMs) are a broad range of non-prescription medicinal products and supplements widely available but generally not considered part of the dominant healthcare system nor conventional medical care [1]. CMs include vitamin and mineral supplements, herbal and botanical medicines, homeopathic preparations, and aromatherapy oils. Most CM products, when manufactured in accordance with Good Manufacturing Practice (GMP) and used appropriately for general health and self-limiting or minor conditions, are considered lower-risk medicines [2]. However, this risk-based approach pays little attention to the risks associated with concurrent use of CM products with other medicines.

Australians are some of the highest users of CM products in the developed world. In 2020, the Australian CM industry yielded sales of AUD 5.69 billion [3]. More broadly, the prevalence of traditional and CM use in the Asia and Western Pacific has attracted public health and policy makers interest regarding how traditional and complementary medicines can be safely integrated to maximize universal health coverage [4]. In Australia, reasons for choosing to use CM products include an overall dissatisfaction with conventional healthcare, a mistrust of pharmaceutical medication, a desire to have more control over health and wellbeing, and an alignment with personal views and traditions of holistic care [5]. While regulation of CM products varies between countries [6], in Australia they are regulated by the Australian Department of Health’s Therapeutic Goods Administration (TGA) with the majority classified as “listed medicines” rather than registered medicines [7]. Listed medicines contain ingredients from a TGA approved list of permissible ingredients. Listed products must be manufactured under GMP. Using TGA-permissible ingredients and manufacturing under GMP are considered indicators of quality and safety [2]. Listed medicines are required to display an “Aust L” number but are not required to have labelling pre-approved before reaching the market and there is no requirement to indicate potential drug–herb interactions included on labelling [7]. In addition, most CMs are not accompanied by consumer medicines information outlining potential drug–CM interactions, despite evidence to support such risks exist.

In 2019, the first study in over a decade to report the prevalence, characteristics, and predictors of Australian CM product use among a representative sample of the Australian adult population was published [8]. This study, which collected data during the year 2017, showed being female, having a higher level of education, living with a chronic health condition and not having private health insurance all predicted CM product use [8]. Findings revealed that 50% of Australians had used a CM product in the previous 12 months (2017), 18% did not disclose their herbal medicine use to their health professionals, and products were mainly self-selected.

Undoubtedly, the last three years of the global pandemic have changed the way many people access healthcare [9]. In Australia, there was an unprecedented increase in telehealth consultations as a primary means for general practitioner services [9]. However, patients who were older reported lower educational qualifications and lower health literacy scores, and lacked access to the Internet reported dissatisfaction with this mode of healthcare delivery [10]. The introduction of electronic prescriptions in Australia in 2020, enabling prescriptions to be sent to patients and pharmacists via email was associated with an overall reduction in patient-specialist consult [11]. Lowered interactions between patients and specialist healthcare professionals reduces the opportunity to discuss and review current medications while increasing the risk of inappropriate and unsafe medication use. Conversely, community pharmacists in Australia represented a direct point of access for patients during the pandemic, thereby positioned at the frontline of healthcare provision and pharmaceutical care [12]. CM products are primarily accessed through pharmacies or retail outlets [13] and common sources of recommendation include general practitioners, pharmacists, store assistants, CM practitioners and self-selection/prescription [9]. The relationship between source of CM product recommendation and concurrent use with other medications has not been elucidated over the last five years or during the COVD-19 pandemic.

Patterns of concurrent use of pharmaceutical medications and certain CM products are associated with pharmacokinetic and pharmacodynamic interactions resulting in adverse events and sub-optimal therapeutic outcomes [14]. However, information regarding patterns of same-day use of CM products with prescription and/or over-the-counter medications have not been examined in Australia or in other countries to date. In addition, no data regarding the prevalence and characteristics of CM use in Australia during the COVID-19 pandemic have been examined to date. Understanding the characteristics and prevalence of CM product use alongside same day medication use during the COVID-19 pandemic can help inform policy and practice with a view to guiding the quality use of medicines. In direct response to this gap and need, the aim of the study reported here was to examine the prevalence and characteristics of CM product use and the prevalence and predictors of same day CM use with prescription and/or over-the-counter medications among the Australian population at a time point during the COVID-19 pandemic.

## 2. Materials and Methods

### 2.1. Study Design

An online cross-sectional survey was distributed to Australian adults (18 and over) via the Qualtrics database. The Human Research Ethics Committee at University of Technology Sydney (#ETH216461) in accordance with the Declaration of Helsinki granted ethical approval for the study.

Individuals were eligible to participate if they were Australian adults aged 18 and over. Representativeness was determined using Chi squared analysis comparing the sample population to National Census data [15]. The sample size is large enough to provide sufficient statistical power for inferential analysis.

#### 2.1.1. Recruitment

Participants were recruited using purposive convenience sampling from a pre-existing database of individuals registered to voluntarily participate in research. Individuals from this database who met the inclusion criteria were invited to participate via email. Recruitment and data collection were conducted between 4 February and 18 February 2022. Prior to completing the survey, informed consent was obtained from each participant once participants had read the information page presented. A small financial incentive (AUD 3–AUD 4) was given to participants by the survey panel company, based on the time taken to complete the survey. They also received redeemable points to contribute towards a voucher, gift card or charity of their choice. The average survey completion time was 36 min.

#### 2.1.2. Measurement

The survey was constructed to include four domains covering demographics, health status, health service and product use, and knowledge mobilization. Survey logic was embedded for additional questions to ensure yes/no questions were followed up where relevant to increase depth of understanding and ensure continuity. This analysis employed data related to demographics, health status and health service and product use.

##### Demographics

Demographic questions included gender, age, residential post code, current level of financial difficulty, highest level of educational qualification, current employment status and present relationship status. Participants were further asked whether they had currently had a Centrelink healthcare card which enables Australian residents on low incomes to access substantial government-subsidies for health services and prescription pharmaceuticals. They were also asked if they had private health insurance and whether it covered ancillary services.

##### Health Status

Participants were asked to indicate if they had been diagnosed with or treated for any of 34 chronic conditions in the previous three years. They were able to indicate ‘other long-term health conditions’ and specify the condition through an open text response option. They were also asked to self-rank their health status on a 5-point scale from ‘excellent’ to ‘poor’ health.

##### Health Service and Product Use

Questions related to health service and product use based on a previous national survey conducted by the research team (Harnett, 2019). Participants were presented with survey items regarding use of a range of prescription-only and over-the-counter medicines, including pharmaceutical and CM products (e.g., vitamins and minerals, Western herbal medicines, Chinese herbal medicines, ingested aromatherapy oils), in the last 12 months and the frequency of use (Never, Rarely, Sometimes, Often). They were then presented with a new survey item to identify the source of prescription or recommendation for each identified medicine, product and treatment (e.g., general practitioner, pharmacists, specialist doctor, CM practitioner). Participants who indicated using any pharmaceutical medicine and any ingested CM—defined as vitamin or mineral, herbal medicine, ingested aromatherapy oil—in the previous 12 months were shown a survey item enquiring about their frequency of using a pharmaceutical and a CM on the same day.

### 2.2. Data Management and Analysis

All survey responses (*n* = 2569) were cleaned to remove duplicate or unreliable responses. New variables were generated to categorize the chronic illness items to broader groups (e.g., cardiovascular conditions, musculoskeletal conditions, respiratory conditions). Frequency of prescription source were also recategorized to combine ‘never’, ‘rarely’ and ‘sometimes’, and to combine ‘always’ and ‘often’. These new categories were identified as ‘infrequent’ and ‘frequent’ use, respectively. Participant age categories were generated to match National Census data and allowed for testing of sample representativeness to the Australian population. Participants were then re-categorized into age groups that permitted better sample distribution between CM product users and non-users. The data were analyzed using Stata V.17. Due to missing data discovered during screening, 218 cases were removed, leaving 2351 participants in the final data set. Chi-square tests were performed to examine associations between categorical variables and CM use. All sociodemographic data were included in logistic regression analysis to determine risk factors for CM use. Chi-square goodness of fit tests were used to compare our sample population with the 2021 Census Data. Statistically significant chi-square results (*p* < 0.05) where then tested using Cramer’s V, to test the strength of the association. Statistically significant chi-square results (*p* < 0.25) were also included in backwards stepwise regression to create a model of best fit to calculate demographic risk factors of CM use, same-day use of CM and prescription medications and sources of CM product recommendations/prescription.

## 3. Results

Table 1 shows a comparison of the sample population to the national population. No major differences were found when comparing proportions of gender age and state of residence. Most participants were female (*n* = 1245) and above the age of 60 (*n* = 707).

### 3.1. Prevalence of CM Product, Prescription and Over-the-Counter Medication Use

Almost half of all participants (*n* = 1167, 49.4%) reported using orally administered CM and 486 participants (20.7%) who reported using both oral CM and prescription and/or over-the-counter medications in the previous 12 months reported high use of oral CM products and prescription and/or over-the-counter medications on the same day. The most used oral CM product was vitamin and/or mineral supplements (*n* = 1055, 44.9%), followed by Chinese herbal medicines (*n* = 121, 5.1%), western herbal medicine (*n* = 114; 4.8%), and internal use of aromatherapy oils (*n* = 38; 1.6%). Prescription medication use was reported by 1472 participants (62.6%) and over-the-counter medications were reported by 1272 participants (54.1%).

### 3.2. Associations between Sociodemographic Characteristics and Oral CM Product Use

Table 2 presents the participants’ sociodemographic characteristics. There was a slightly greater proportion of participants who identified as female (53.0%), and the majority reported having a trade certificate or equivalent (32.3%) or a university qualification (31.5%). Most participants’ relationship status was married or *de facto* (*n* = 1375).

Many participants also reported having chronic health issues (*n* = 1533; 65.2%). These included mental health and psychiatric disorders (*n =* 669, 28.5%), cardiovascular conditions (*n* = 500; 21.3%), female reproductive conditions (*n* = 314; 13.4%), respiratory conditions (*n* = 314; 13.4%), musculoskeletal disorders (*n =* 277; 11.8%), gastrointestinal disorders (*n* = 274; 11.7%), diabetes (*n* = 258; 11.0%), migraine (*n* = 218, 9.3%), COVID-19 (*n* = 178; 7.6%), benign or malignant cancer (*n* = 168; 7.1%), idiopathic fatigue disorders (*n* = 81; 3.4%), autoimmune disorders (*n* = 79; 3.4%) and male reproductive disorders (*n* = 69; 2.9%).

χ2 tests of association found a statistically significant relationship (*p* < 0.05) between oral CM product use and participant gender, age, relationship status, highest qualification, private health insurance cover, and incidence of chronic health diagnoses, when compared to non-users (Table 2).

### 3.3. Predictors of Oral Use of CM Products and Same-Day Use of These Products with Pharmaceutical Medicines

The Model of best fit for oral CM product use was statistically significant (χ2(14) = 184.73, *p* < 0.001; *n* = 2351) and correctly classified 60.66% of cases. As seen in Table 3, there was an increased likelihood of using oral CM products among females (OR 1.46) compared to males. Individuals with a post-high school qualification were also more likely to use an oral CM compared to participants who had completed year 10 or less (OR 1.57–2.24). Participants with health insurance cover similarly had an increased likelihood of using oral CM compared to those without health insurance cover. There was also an increased likelihood of oral CM product use among individuals who self-reported diagnosis with a male reproductive condition (OR 2.09), a musculoskeletal disorder (OR 1.97), a cardiovascular condition (OR 1.64), a gastrointestinal disorder (OR 1.55), cancer (OR 1.41), or a mental health or psychiatric condition (OR 1.27), compared to participants that did not report diagnosis with these conditions.

### 3.4. Sources of Recommendation by Same-Day Use

Table 4 reports the results of a chi-square tests of association between sources of recommendation of CM product use and the frequency of same-day use of CM and pharmaceutical medication. The most common source of recommendation of oral CM products was self-selection (*n* = 588; 45.9%), followed by GP (*n* = 488; 38.1%), family or friend (*n* = 207; 16.2%), pharmacist (*n* = 187; 14.6%), specialist doctor (*n* = 167; 13.0%), CM practitioner (*n* = 100; 7.8%), pharmacy or health food store assistant (*n* = 80; 6.2%), and hospital doctor (*n* = 63; 5.0%). The incidence of recommendation of oral CM products by all types of medical doctors (GP, specialist doctor, hospital doctor) was reported by 601 participants (46.9%).

The most common source of recommendation for vitamin and mineral supplements, the most highly used oral CM product, was self-selection (*n* = 430), followed by GP (*n* = 353) (see Table 4). Same-day use of CM products and prescription and/or over-the-counter medications were reported as occurring frequently by more than half of participants using vitamins and/or mineral supplements (54.6%) but not for users of other types of oral CM products. This proportion was significantly greater among vitamin and/or mineral supplement users who had been recommended their use by a GP (61.5%; *p* < 0.001) or specialist doctor (70.6%; *p* = 0.001) and was less common among users who self-selected (51.0%; *p =* 0.03) or were recommended to use vitamins and/or minerals by a family member or friend (46.0%; *p* = 0.02). The lowest proportion of infrequent use of oral CM products and prescription and/or over-the-counter medications on the same day was reported by participants who used internal aromatherapy oils (36.8%).

### 3.5. Predictors of Same-Day Use of Oral CM Products with Prescription and/or Over-the-Counter Medications

The model of best fit for same-day use of orally administered CM and prescription medications was statistically significant (χ2(14) = 189.42, *p* < 0.001; *n* = 908) and correctly classified 70.59% of cases. As seen in Table 5, individuals with ‘very good’ general health were less likely (OR 0.48) than those with ‘excellent’ health to use CM and pharmaceutical medicines on the same day. Individuals who were 35 years and over were significantly more likely (OR 1.80–5.81) than those aged 18–24 years to report same-day use of oral CM and prescription medications, with this likelihood increasing for older participants. Individuals with a university degree or higher were less likely than those who completed only year 10 or less to use CM and prescription medicines on the same day (OR 0.53). Participants with a cardiovascular condition (OR 1.82) or a musculoskeletal disorder (OR 1.70) were also more likely to use oral CM on the same day as a prescription and over-the-counter medications, compared to individuals who did not report diagnosis with those conditions.

## 4. Discussion

This study is the first to investigate same-day use of orally administered CM and prescription and over-the-counter medications in a broadly representative sample of Australian adults. It is also the first study to report this use during the COVID-19 pandemic. We identified that 49.4% of participants used CM products in 2022 during the COVID-19 pandemic. This is similar to the prevalence of use during 2017 amongst an Australian population [9]. The most frequently used CM products were vitamin and mineral supplements (44.9%) again comparable to the 47% of Australians who reported using this category of CM products in 2017 [9,16]. Predictors of overall CM product use in our study remained consistent with previous reports with gender (female), a higher education status, and living with a chronic condition identified as predicters. Conversely, the current study found that having private health insurance increased the likelihood of CM use in 2022, which contradicts the 2017 study reporting a converse relation that not having private health insurance predicted CM use [9].

The main finding in our study is that >50% of the Australian adults using CM products in our sample always or often took the product on the same day as other medicines, and those greater than 65 years of age were five times more likely that those aged between 18 and 24 years of age to do so. This finding is supported by a similar study conducted in the state of South Australia in 2019 that found 40% of older South Australians took their prescription and/or over-the-counter (OTC) medicines with CMs on the same day, and same-day use increased with age [17]. Similar findings were reported in 2006 with 50% of older South Australian residents using multiple medicines including CM products on the same day [18]. Older people are more likely to have chronic conditions and to be taking multiple medications increasing the risks of drug–drug and potential drug–herb-nutrient interactions [19,20]. There is a real opportunity for those providing older people’s healthcare to enquire about all medicines used, access drug interaction information resources and document all medicines use during medicine reconciliation and home medicine reviews. Importantly, this present study identified that other potentially vulnerable populations including those with chronic cardiovascular and musculoskeletal conditions are at highest risk of CM–drug interactions associated with same-day use. These findings regarding same-day use, may be attributed to the increased likelihood of suffering from a chronic illness and engaging in pharmaceutical medicine use as age increases [21]. This finding supports the need for all healthcare professionals to not only ask their patients about CM product use, but to also be informed regarding common drug–herb interactions that may occur, especially in these higher-risk populations.

In addition to older age, a lower education level also increased the likelihood of same-day use in comparison to those with tertiary education in our study. In contrast, the South Australian study involving older people reported an increase in same day among those with a higher education level and be in excellent health [17]. There remains a real opportunity for targeted education regarding the concurrent use of CMs and other medications of these sub-populations and the healthcare professionals they consult. Further, given the prevalence of concurrent use and lack of data regarding the benefits and risks of various combinations of CM and pharmaceutical medicines, research and education focused on this area is also warranted.

The prevalence of self-prescribing all CM products, predominantly vitamin and mineral supplements, in our study remained similar to that identified in the 2017 study [9]. While this limits the potential for engagement with healthcare professionals in a clinical setting, there is nevertheless a real opportunity for pharmacy staff directly involved in the provision of medicines, including CMs, to provide medicines counselling. In our 2022 data, second to self-prescription, GPs were the most common source of CM product recommendations compared to 2017 when it was reported to be pharmacists [9]). A possible explanation for this could be increased opportunity to discuss CM use with the use of tele-health GP consultations during the pandemic. Our study also shows self-prescribing is the most frequent source of recommendation for those engaging in same-day use of CM and prescription medicine further supporting the need for research in this area to establish whether the concurrent use is well informed, provides benefits or increases the risk and incidence of adverse treatment outcomes.

This study raises important questions about the potential for drug–herb interactions among Australians that are more likely to engage in same-day use of CM products and pharmaceutical medicines and likely relevance to day-to-day practice, considerations relevant for all healthcare professionals. Many oral CM products contain pharmacologically active constituents that can interact with prescription medicines through pharmacokinetic and pharmacodynamic mechanisms [22]. There are various mechanisms that influence the efficacy, activity, therapeutic window, and subsequent safety of prescription medication. A common mechanism is via inhibition or induction of drug metabolizing cytochrome (CYP) P450 enzymes [22]. Via these mechanisms, CM products can alter blood concentrations of prescription medications, leading to toxic effects due to prolonged exposure, or compromising drug efficacy by reducing concentration [22]. St John’s Wort (*Hypericum perforatum*) has been studied extensively regarding its CYP 3A4 and p-glycoprotein inducing effects [23]. These pharmacokinetic interactions are associated with reducing the effectiveness of a range of medications used in the management of chronic conditions including immunosuppressants, antiretrovirals, anticancer medications, antiarrhythmics, anticoagulants and oral contraceptives. Interactions between CM products and medications commonly used in chronic conditions include case reports identifying ginkgo biloba, garlic (*Allium sativum*), turmeric (*Curcuma longa*) and ginseng (*Panax ginseng*), and kava kava (*Piper methysticum)* [24]. While our study reports the prevalence of same-day use of CM and pharmaceutical medications, we did not elucidate details regarding medicine names and combinations and there remains a lack of evidence regarding the interaction potential of many drug-pairs [25]. This highlights a need to drive research on CM interactions and develop publicly available, up-to-date databases containing interaction information [26] and increase CM health literacy. Identifying interactions will facilitate improved CM product labelling to include important information regarding contraindications and potential drug–drug interactions, for easy reference by healthcare professionals and consumers [26]. Thus, those engaging in and those recommending CM use will have greater access to key safety information and self-prescribing consumers can make informed decisions.

While the COVID-19 pandemic changed the way Australians accessed and interacted with the healthcare system, for the most part the patterns and predictors of CM use appear to have remained consistent with pre-COVID-19 findings [9]. Our study finding that GPs were the most common healthcare professional to be recommending CM products was surprising given the impact of COVID-19 restrictions but maybe associated with increase in tele-health consultations [27,28]. The seemingly small impact of COVID-19 on the prevalence of CM product use by Australians indicates that Australians continue to value the role of these products in their healthcare and their accessibility was not compromised during that time.

Limitations of the study must be considered. Due to the self-report nature of the survey, the study is susceptible to potential recall or responder bias, particularly as participants reported on items from the previous 12 months. However, the survey included validated instruments used commonly in health services research. In addition, the large, representative sample affords generalisability of findings to the Australian adult population, increasing their value for researchers, policy makers and health-professionals. The study only examined CM products and pharmaceuticals as broad categories, and as such the actual risks associated with specific drug–herb and drug–nutrient interactions cannot be fully determined. Future research must build on the findings of this study to focus on relevant illness populations (e.g., individuals with cardiovascular and musculoskeletal conditions) and the types of CM products and pharmaceuticals of greatest interest and use in those populations. The finding that the prevalence of CM product use by Australians remained consistent since the data collected in 2017 despite a global pandemic was not expected. However, this result is likely associated with the study design, i.e., the survey asked about the previous 12 months (2021) of use and did not capture CM use earlier in the pandemic prior to the widespread availability of vaccines and changes to lockdown measures.

## 5. Conclusions

The use of CM products by Australians has remained consistent over the last five-year period despite a global pandemic. The same-day use of CM products with prescription and over-the-counter medications among older people and those with chronic conditions requires further consideration by researchers, practitioners and policymakers in order to help prevent harm and foster the quality use of medicines.

## Figures and Tables

**Table 1 nutrients-15-00327-t001:** Sociodemographic characteristics of survey participants (*n* = 2351) compared with national data from the 2021 National Census.

Characteristics	Survey Participants (*n =* 2351)		National Census Data (2021)		*p*
	*n*	%	*n*	%	
Gender					
Male	1081	46	9,828,089	49.04	0.617
Female	1245	53	10,209,528	50.96	
Other	25	1.1	-	-	
Age					
18–19	92	3.9	610,945	3.0	0.616
20–29	512	21.7	3,617,689	18.1	
30–39	427	18.2	3,757,954	18.8	
40–49	390	16.6	3,296,519	16.5	
50–59	223	9.5	3,120,900	15.6	
60 and over	707	30.1	5,633,610	28.1	
State of residence					
New South Wales/Australian Capital Territory	763	32.4	6,718,095	33.5	0.917
Victoria	574	24.4	5,261,500	26.3	
Queensland	477	20.3	3,986,990	19.9	
South Australia/Northern Territory	203	8.6	1,585,085	7.9	
Western Australia	247	10.5	2,054,078	10.3	
Tasmania	87	3.7	428,097	2.1	

**Table 2 nutrients-15-00327-t002:** Association between participant demographic characteristics and use of oral CM products.

	Total (*n =* 2351)		Oral CM Product Use (*n =* 1167)		*p*
	*n*	%	*n*	%	
Gender					
Female	1245	53.0	656	56.2	<0.001
Male	1081	46.0	496	42.5	
Non-binary/Other	25	1.1	15	1.3	
Age					
18–24	350	14.9	145	12.4	0.01
25–34	461	19.6	228	19.5	
35–44	449	19.1	238	20.4	
45–54	269	11.4	127	10.9	
55–64	264	11.2	136	11.7	
*65 and over*	558	23.7	293	25.1	
Relationship Status					
Never Married	729	31.0	331	28.4	0.02
Married/De Facto	1375	58.5	710	60.8	
Separated/Divorced/Widowed	247	10.5	126	10.8	
Qualification					
Year 10 or less	341	14.5	134	11.5	0.001
Year 12	510	21.7	222	19.0	
Trade/Apprenticeship/Certificate/Diploma	760	32.3	383	32.8	
University Degree or Higher	740	31.5	428	36.7	
Employment Status					
Full-Time Work	770	32.8	385	33.0	0.06
Part-Time or Casual	547	23.3	293	25.1	
Looking for Work/Not in the paid workforce	1034	44.0	489	41.9	
State					
New South Wales	704	29.9	366	31.4	0.5
Victoria	574	24.4	284	24.3	
Queensland	477	20.3	238	20.4	
Western Australia	247	10.5	114	9.8	
Other states and territories	349	14.8	165	14.1	
General health					
Excellent	175	7.4	80	6.9	0.3
Very good	595	25.3	315	27.0	
Good	940	40.0	460	39.4	
Fair	497	21.1	238	20.4	
Poor	144	6.1	74	6.3	
Private health insurance cover	1132	48.1	624	53.5	<0.001
Healthcare card	1343	57.1	673	57.7	0.6
Chronic health diagnosis	1533	65.2	834	71.5	<0.001

**Table 3 nutrients-15-00327-t003:** Characteristics predicting oral CM product use.

	Odds Ratio	95% CI	*p*
**Gender**			
*Male*	Ref	-	-
*Female*	1.46	[1.22, 1.74]	0.000
*Non-binary/Other*	2.07	[0.89, 4.80]	0.091
**Qualification**			
*Year 10 or less*	Ref	-	-
*Year 12*	1.27	[0.94, 1.72]	0.114
*Trade/Apprenticeship/Certificate/Diploma*	1.57	[1.19, 2.07]	0.001
*University Degree or Higher*	2.24	[1.67, 3.01]	0.000
**Employment Status**			
*Full-Time Work*	Ref	-	-
*Part-Time or Casual*	1.17	[0.93, 1.48]	0.174
*Looking for Work/Not seeking labor*	0.89	[0.72, 1.10]	0.290
**Health insurance cover**	1.40	[1.17, 1.66]	<0.001
**Cancer (benign/malignant)**	1.41	[0.99, 2.00]	0.054
**Cardiovascular conditions**	1.64	[1.30, 2.06]	<0.001
**Musculoskeletal disorders**	1.97	[1.47, 2.63]	<0.001
**Gastrointestinal conditions**	1.55	[1.17, 2.06]	0.002
**Mental health/Psychiatric conditions**	1.27	[1.05, 1.55]	0.014
**Male Reproductive conditions**	2.09	[1.18, 3.68]	0.011

**Table 4 nutrients-15-00327-t004:** Sources of Recommendation for Participants Using Orally Administered CM Products.

Source of Recommendation of CM Product	Same-Day Use of CM and Pharmaceutical				
	Infrequent		Frequent		*p*
	*n*	%	*n*	%	
Chinese herbal medicine (*n =* 78)	42	53.9	36	46.2	
GP (*n =* 16)	9	56.3	7	43.8	0.8
Specialist doctor (*n =* 8)	6	75.0	2	25.0	0.2
Hospital doctor (*n =* 3)	3	100.0	0	0.0	0.09
Pharmacist (*n =* 10)	8	80.0	2	20.0	0.08
Pharmacy or Health food store assistant (*n =* 4)	4	100.0	0	0.0	0.06
CM practitioner (*n =* 13)	6	46.2	7	53.9	0.5
Self-selected (*n =* 23)	11	47.8	12	52.2	0.5
Referred by family member (*n =* 24)	13	54.2	11	45.8	0.9
Vitamins and/or mineral supplements (*n =* 944)	429	45.4	515	54.6	
GP (*n =* 382)	147	38.5	235	61.5	<0.001
Specialist doctor (*n =* 102)	30	29.4	72	70.6	0.001
Hospital doctor (*n =* 31)	12	38.7	19	61.3	0.4
Pharmacist (*n =* 110)	58	52.7	52	47.3	0.1
Pharmacy or Health food store assistant (*n =* 41)	19	46.3	22	53.7	0.9
CM practitioner (*n =* 44)	20	45.5	24	54.6	1.0
Self-selected (*n =* 467)	229	49.0	238	51.0	0.03
Referred by family member (*n =* 148)	80	54.1	68	46.0	0.02
Western herbal medicine (*n =* 92)	55	59.8	37	40.2	
GP (*n =* 28)	17	60.7	11	39.3	0.9
Specialist doctor (*n =* 22)	13	59.1	9	40.9	0.9
Hospital doctor (*n =* 15)	10	66.7	5	33.3	0.5
Pharmacist (*n =* 21)	10	47.6	11	52.4	0.2
Pharmacy or Health food store assistant (*n =* 12)	8	66.7	4	33.3	0.6
CM practitioner (*n =* 18)	11	61.1	7	38.9	0.9
Self-selected (*n =* 28)	15	53.6	13	46.4	0.4
Referred by family member (*n =* 14)	10	71.4	4	28.6	0.3
Internal aromatherapy oils (*n =* 19)	12	63.2	7	36.8	
GP (*n =* 5)	2	40.0	3	60.0	0.3
Specialist doctor (*n =* 3)	2	66.7	1	36.8	1.0
Hospital doctor (*n =* 3)	0	0.0	3	100.0	0.04
Pharmacist (*n =* 2)	1	50.0	1	50.0	1.0
Pharmacy or Health food store assistant (*n =* 1)	0	0.0	1	100.0	0.4
CM practitioner (*n =* 2)	2	100.0	0	0.0	0.5
Self-selected (*n =* 7)	4	57.1	3	42.9	1.0
Referred by family member (*n =* 1)	1	100.0	0	0.0	1.0

**Table 5 nutrients-15-00327-t005:** Participant characteristics predicting same-day use of oral CM products with prescription and/or over-the-counter medications.

Participant Characteristics	Odds Ratio	95% CI	*p*
General Health			
*Excellent*	Ref	-	-
*Very good*	0.48	[0.24, 0.93]	0.029
*Good*	0.65	[0.34, 1.24]	0.194
*Fair*	0.67	[0.34, 1.34]	0.259
*Poor*	2.03	[0.85, 4.87]	0.113
**Age**			
*18–24*	Ref	-	-
*25–34*	1.30	[0.73, 2.29]	0.372
*35–44*	1.80	[1.03, 3.14]	0.040
*45–54*	1.96	[1.06, 3.62]	0.033
*55–64*	2.36	[1.28, 4.35]	0.006
*65 and over*	5.81	[3.30,10.24]	0.000
**Qualification**			
*Year 10 or less*	Ref	-	-
*Year 12*	0.89	[0.48, 1.65]	0.721
*Trade/Apprenticeship/Certificate/Diploma*	0.66	[0.38, 1.14]	0.135
*University Degree or Higher*	0.53	[0.30, 0.93]	0.026
**Cardiovascular conditions**	1.82	[1.26, 2.61]	0.000
**Musculoskeletal disorders**	1.70	[1.10, 2.62]	0.016

## Data Availability

The study data analyzed for this paper are not publicly available but requests for access can be made to the corresponding authors.

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
