# Peer review of "Prevalence and Characteristics of Australians’ Complementary Medicine Product Use, and Concurrent Use with Prescription and Over-the-Counter Medications—A Cross Sectional Study"

_nutrients, 2023, doi:10.3390/nu15020327_

Round 1
Reviewer 1 Report
11. This paper is well-written, with very clear prose and tables that illustrate the findings. It follows in a tradition of papers reporting complementary and alternative medicine use gathered from large scale surveys in Europe, North America, and Australia. That said, there are a number of corrections and clarifications needed.
22. The introduction needs clarification. The rationale for the paper—that no one has reported use of CM in the past 5 years--is very confusing. There are papers cited (e.g., reference 9) that are recent, but do not give the date of data collection. Need to distinguish between when a study was CONDUCTED and when it was PUBLISHED, especially considering that the authors make statements about the need for more recent studies.
33. The rationale that one should be concerned about CM because it can interact with prescription medicine seems to have relatively limited support, according to the introduction. The authors cite a 2017 paper. There appear to be no data on the number, type, and severity of such interactions (though some is cited in the discussion). Therefore, this paper, which documents concurrent CM and prescription medicine use, is a start to either supporting or debunking this argument. Such work is needed.
44. Please specify the amount of the financial incentive given to participants.
55. For an international audience, please explain what a Centrelink healthcare care is. Line 124.
66. The authors have a very broad definition of complementary medicine and they do not distinguish that prescribed by a conventional healthcare provider (e.g., calcium for bone density) from that self-prescribed by the respondent. In the end, they lump all forms of vitamins, minerals, herbs, etc. together. This is an unmentioned weakness of the paper.
77. The Discussion stresses that the study is first to do several things in Australian adults. The authors should comment on whether there are studies outside Australia that are comparable. It is not clear why this paper restricts comparisons to only Australia.
88. Although the data were gathered on different types of CM, the main analyses looking at concurrent use lumps all CM together and lumps all prescription drug use together. The authors then conclude that the study raises important questions about drug-herb interactions. This reviewer would argue that it tells us nothing specific about herbs and, because such data were collected, why can some of the questions not be answered with the data collected in this study? The interactions cited—St John’s Wort, gingko with prescriptions medicine—have been in the literature for years—why did the authors not gather data specifically on them for this study? What a wasted opportunity!
99. Sentence fragment—lines 343-4.
110. Overall, this is a well written, but very disappointing study. It was well conducted and analyzed, but the authors did not really collect specific enough data to answer the questions they pose. Yes, science is incremental, but the incremental gain shown from this paper is woefully small.
Author Response
Please find our point-by-point response to your comments attached.

Reviewer 2 Report
Joanna et al investigate the epidemiology of complementary and medicine product use in Australians by real-world setting. A total of 2351 adult were enrolled for surveying. There are some suggestion:
1. Authors define chi-square results p<0.025 define as statistically significantly difference. However, there is provided reason for define 0.025.
2.Authors should address the multiplicity problem. There were variables have questioned about multiplicity group based on lack of power. Bonferroni adjustments can consider for testing.
3.age group is hard to understanding in table 1 and table 2. I think you can use same criteria for each group (eg. 10 years old ).
4.Table 2 is hard for reading, due to there was no providing non-users, please add in table 2.
5. Health status is hard to define from current data. Please provide the definition of excellent health to poor health.
6. It is hard to know the impact of COVID-19. Authors have to explain in detail.
The results are interesting and provide some reference value for other countries. Also, it is well-written and the sample-size is enough.
Author Response

(The authors gave the same response as above.)

Round 2
Reviewer 1 Report
This paper has been revised to address the reviewer concerns. The changes improve the paper and, in particular, highlight the need for future, more detailed research. The changes also highlight the contribution of this particular piece of research to the overall complementary medicine research field.